# Land Cover Extraction in the Typical Black Soil Region of Northeast China Using High-Resolution Remote Sensing Imagery

**Binbin Ding [1], Jianlin Tian [1,\*,†], Yong Wang [2,\*,†]** and **Ting Zeng [1]**

[1]   College of Civil Engineering and Architecture, Jishou University, Zhangjiajie 427000, China; 2020700501@stu.jsu.edu.cn (B.D.); 2021700530@stu.jsu.edu.cn (T.Z.)

[2]   State Key Laboratory of Resources and Environmental Information System, Institute of Geographical Sciences and Natural Resources Research, Chinese Academy of Sciences, Beijing 100101, China

\*   Correspondence: 007319@jsu.edu.cn (J.T.); wangy@igsnrr.ac.cn (Y.W.); Tel.: +86-135-7446-3606 (J.T.); +86-10-6488-8179 (Y.W.)

†   These authors contributed equally to this work.

**Abstract:** The black soil region of Northeast China is one of the most fertile soil areas in the world and serves as a crucial grain-producing region in China. However, excessive development and improper utilization have led to severe land use issues. Conducting land cover extraction in this region can provide essential data support for monitoring and managing natural resources effectively. This article utilizes GF-6 remote sensing imagery as the data source and adopts the U-Net model as the backbone network. By incorporating residual modules and adjusting the convolution kernel size, a high-precision land cover extraction model called RAT-UNet is developed. Taking Qiqihar City as an example, the RAT-UNet model is applied to extract land cover information. The results are as follows: (1) The RAT-UNet model achieves high accuracy in land cover extraction, with the following accuracies for different land types: cropland (95.11%), forestland (93.61%), grassland (68.41%), water bodies (94.67%), residential land (89.40%), and unused land (87.25%). (2) The land cover extraction performance of the RAT-UNet model is superior to DeepLabV3, U-Net, SegNet, and LinkNet34 models. This research outcome provides methodological support for the intelligent and high-precision extraction of land cover information and also offers timely data for Qiqihar city's land use planning.

**Keywords:** U-Net model; RAT-UNet model; land cover extraction; GF-6; Qiqihar City

## 1. Introduction

The black soil region of Northeast China is one of the four major black soil regions in the world [1]. It includes Heilongjiang Province, Jilin Province, the northeastern part of Liaoning Province, and the "Eastern Four Leagues" region of Inner Mongolia. The total area of black soil in this region is about 1.09 million square kilometers, with 185,300 square kilometers being typical black soil [2]. The typical black soil area in Heilongjiang Province accounts for 56.1% of the total typical black soil area in the black soil region of Northeast China. Among them, Qiqihar City in the western part of Heilongjiang Province is located in the heartland of the Songnen Plain black soil area and is often referred to as the "Black Soil Pearl." It serves as a representative example of the black soil habitat for giant pandas. In recent years, due to the continuous progress of agricultural development and urbanization, Qiqihar City has faced serious land use issues. In line with the objective of protecting the black soil, the region has undertaken a geographic information survey to optimize the utilization and direction of regional natural resources. This effort is beneficial for implementing the requirements for black soil preservation and the "storing grain in the land and storing grain in technology" strategic approach.

The extraction of land cover in the black soil region mainly relies on traditional field survey methods and remote sensing-based extraction methods. The field survey method is based on collected data from the black soil region, which is used to analyze the current land use status and identify existing issues, proposing corresponding strategies. However, this approach is affected by various factors such as the size of the survey area, topography, and weather conditions, which significantly limits the scope of the survey and makes it challenging to meet the demands for large-scale land cover extraction in the black soil region. With the rapid advancement of remote sensing technology, remote sensing satellite imagery has become the primary data source for land cover extraction. Some scholars have utilized multi-temporal remote sensing data from the black soil region and achieved land cover classification through visual interpretation methods, providing strategies for land planning and management in the black soil region. Other researchers have constructed a multi-dimensional index system for land cover classification in the black soil region, using spatial overlay techniques to analyze the quantity and spatial distribution characteristics of various land cover categories. This approach has laid a solid data foundation for the rational utilization and conservation of black soil [3]. Although remote sensing-based methods for land cover extraction in the black soil region have enabled large-scale extraction, the results can be influenced by factors such as expertise and subjectivity. Moreover, these methods still require substantial human and material resources, making it challenging to meet the timely land cover extraction demands of the black soil region.

With the development of artificial intelligence technology, land cover extraction has encountered new opportunities, and AI-based land cover extraction has emerged as a novel approach. Initially, machine learning methods based on shallow feature extraction caught the attention of many scholars. They focused on comparing different machine learning algorithms [4–8]. Subsequently, in order to further improve the accuracy of land cover extraction, scholars became enthusiastic about research on the improvement of machine learning algorithms [9,10]. Although land cover extraction based on machine learning algorithms has shown improvements in both accuracy and efficiency compared to traditional field survey methods and remote sensing-based extraction methods, analyzing the various methods used over the years, it is evident that this approach heavily relies on data quality and is susceptible to factors such as sample size and parameter settings. Therefore, land cover extraction still presents certain challenges. Subsequently, deep learning algorithms based on deep feature extraction gained prominence due to their ability to effectively address the problem of machine learning algorithms' inability to adapt to target feature variations after feature engineering [11]. Moreover, these algorithms can extract more complex features, making the extraction process more intelligent and leaving an unforgettable impression in many fields [12]. Numerous studies have utilized convolutional neural networks (CNN) as the foundation, such as land cover and crop classification in multiple temporal scenes [13], agricultural land use classification [14], urban feature classification [15], land cover classification based on satellite imagery [16], and more. These studies have demonstrated that CNN is a feasible tool for solving remote sensing data segmentation and target recognition tasks. However, it is important to note that CNN as a representative of convolutional neural networks, suffers from the issue of losing local information, leading to blurry object boundaries and difficulties in accurately discerning pixel categories from abstract features, resulting in imprecise segmentation. To address these limitations, Long et al. made improvements to the CNN architecture by replacing the fully connected layers with convolutional layers, leading to the proposal of Fully Convolutional Networks (FCN) capable of achieving pixel-level classification [17]. To obtain deeper semantic information, several more advanced networks have been developed [18], including FCN-8s [19], DeepLabV3+ [20,21], U-Net [22], and others. These networks aim to enhance the segmentation accuracy and capture more detailed features within the images.

U-Net model, as a typical end-to-end architecture, has gained significant attention and popularity in various fields. Initially recognized for its outstanding performance in medical image semantic segmentation [23], the U-Net model has been subsequently

widely applied in various domains, such as land cover extraction, vegetation extraction, road extraction, and more [24,25]. The U-Net model employs a contracting path with convolutional and pooling layers for feature extraction and may encounter the issue of model degradation as the network gets deeper, leading to limited feature information extraction. To improve the accuracy of land cover extraction, Singh et al. attempted to add Dropout layers after each convolutional layer in the U-Net model. The experimental results showed that incorporating Dropout layers effectively addressed the weight-sharing problem, but the learning process still suffered from overfitting [26]. Another attempt by Li et al. introduced dual attention modules to enhance the extraction performance of different land cover types, but this approach increased the computational complexity and lacked completeness in detecting large-scale land cover features [27]. Several researchers have also explored the incorporation of residual modules and dilated spatial pyramid pooling into the network architecture, applying the improved models to urban land cover extraction and high spatial resolution image classification. These studies demonstrated the effectiveness of residual modules and dilated spatial pyramid pooling in enhancing land cover extraction performance. However, these models exhibited higher accuracy only on public datasets and showed relatively lower performance on personally constructed training datasets, leaving room for improvement when dealing with individually constructed datasets [28–30].

Currently, land cover extraction research in the black soil region of Northeast China heavily relies on traditional methods, which consume significant human and material resources. These methods struggle to meet the high-precision and timeliness requirements for land cover extraction in the black soil region. To address this issue, constructing an intelligent and high-precision land cover extraction model based on deep learning algorithms can effectively enhance the accuracy and efficiency of obtaining land cover information. Additionally, such a model can provide timely data for natural resource planning and management in the black soil region. In this study, we propose to build an intelligent land cover extraction model based on deep learning methods. Drawing inspiration from existing land cover extraction models and leveraging a personally constructed sample library for the black soil region, we will utilize the U-Net model as a foundation and introduce residual modules and adjust the convolutional kernel size in the model, creating the RAT-UNet model. Subsequently, we will apply the RAT-UNet model to land cover extraction in Qiqihar City to verify its effectiveness, offering a novel approach for land cover extraction research in the black soil region of Northeast China.

## 2. Materials and Methods

### 2.1. Study Area

Qiqihar City is located in the western part of Heilongjiang Province, China (45° N~48° N, 122° E~126° E). It covers a vast area and is currently administratively divided into 7 districts and 9 counties. The total land area of the city is approximately 42,300 square kilometers. The region has a temperate continental climate with four distinct seasons. Qiqihar City is rich in black soil resources, including six types of soil: black soil, black calcareous soil, meadow soil, albic soil, dark brown soil, and brown soil [31]. The typical area of black soil cropland is 7720 square kilometers, making it the fourth largest grain-producing city in China. The dominant landforms in Qiqihar City are plains and hills [32]. The land use types include cultivated land, forest land, grassland, water bodies, and residential land. In recent years, with the continuous development of agriculture and urbanization, as well as the intensive and overloaded cultivation practices in the black soil region, ecological and land degradation issues have become increasingly prominent in the area.The administrative division map of the study area is shown in Figure 1.

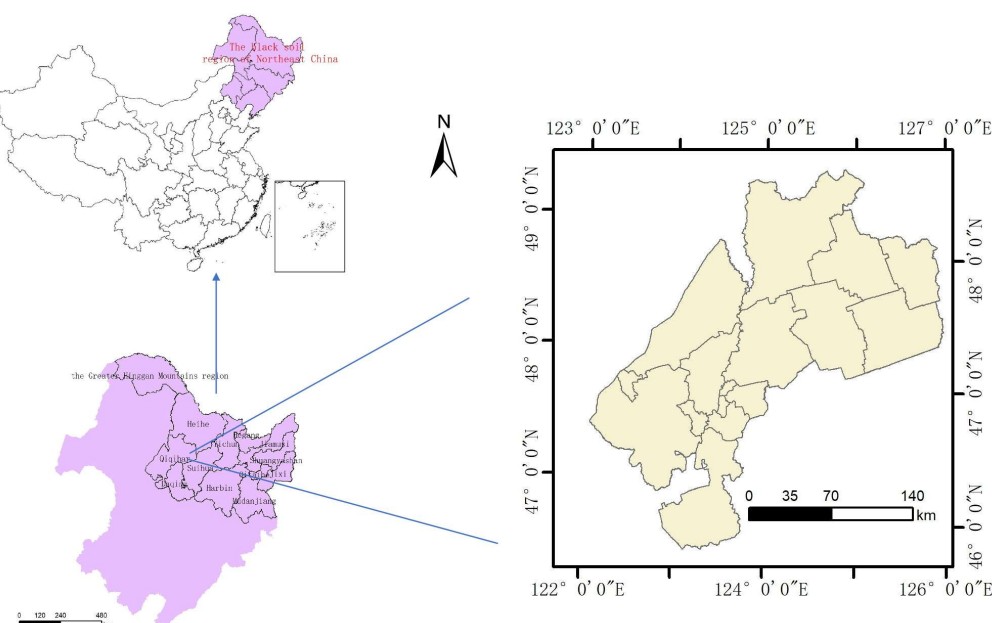

**Figure 1.** Administrative division map of Qiqihar City.

## 2.2. Dataset

### 2.2.1. Data Source and Preprocessing

This article mainly uses the domestic High-Resolution 6 (GF-6) multispectral satellite imagery as the primary research data, downloaded from the China Center for Resources Satellite Data and Application (https://www.cresda.com/zgzywxyyzx/index.html, accessed on 1 December 2021). The data selection was based on the criteria of cloud cover less than 2% and includes 9 scenes of remote sensing satellite imagery covering the entire growing season (May to October). The GF-6 satellite's PMS (Panchromatic Multispectral Sensor) sensor has one panchromatic band with a spatial resolution of 2 m and four multispectral bands with a spatial resolution of 8 m. The observation swath is 90 km, and the revisit period has been reduced from 4 days for GF-1 to 2 days for GF-6. It possesses characteristics such as high resolution, wide coverage, high quality, and efficient imaging.

Based on the spectral characteristics of the land cover in the black soil region, a combination of blue, green, and red bands is used to display the true colors of the ground objects, allowing for effective observation of various land cover categories. The GF-6 multispectral satellite imagery is preprocessed using ENVI 5.3 software. First, the Radiometric Calibration tool is used to perform radiometric calibration on the GF-6 panchromatic and multispectral imagery, eliminating sensor-related errors. This step ensures accurate radiometric values in the imagery. Second, the FLAASH Atmospheric Correction tool is applied to the multispectral imagery to correct for atmospheric effects such as scattering and absorption. This correction helps remove errors caused by atmospheric conditions and illumination variations. Next, the RPC Orthorectification Workflow tool is utilized to perform orthorectification on both the multispectral and panchromatic imagery. This process corrects geometric distortions in the images, aligning them with the Earth's surface and improving their spatial accuracy. Finally, the NNDiffusePanSharpening fusion tool is employed to merge the panchromatic and multispectral imagery. This fusion technique combines the rich spectral information from the multispectral bands with the high spatial resolution of the panchromatic band. The result is a high-resolution multispectral satellite image with a spatial resolution of 2 m, effectively utilizing both the spectral and spatial details of the imagery [33]. The pre-processing workflow for the remote sensing satellite imagery in the study area is illustrated in Figure 2.

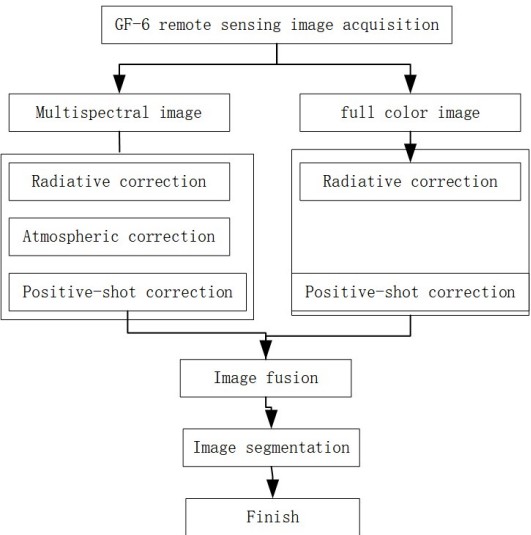

**Figure 2.** Flow chart of GF-6 image pre-processing.

### 2.2.2. Construction of the Land Object Sample Library

Taking into account the nature of black soil and the natural resource characteristics of Qiqihar City and referring to the classification standards of the China National Land Use and Land Cover Dataset (CNLUCC) for multiple time periods [34], a land object sample library for Qiqihar City was constructed. The sample library includes six land object types: cropland, forestland, grassland, water bodies, residential land, and unused land.

First, based on the remote sensing mechanism, the images were initially classified by calculating the Normalized Difference Vegetation Index (NDVI) and Normalized Difference Water Index (NDWI) to initially extract vegetation and water bodies. NDVI is a commonly used vegetation index, which distinguishes vegetation from non-vegetation areas by mathematically transforming the differences between red and near-infrared light. On the other hand, the NDWI index uses mathematical transformations of the differences between green and near-infrared light to effectively suppress vegetation information while highlighting water body information. The formulas for calculating NDVI and NDWI are as follows:

$$NDVI = (B4 - B3)/(B4 + B3) \tag{1}$$

In the formulas, B4 refers to the near-infrared band, and B3 refers to the red band.

$$NDWI = (B2 - B4)/(B2 + B4) \tag{2}$$

In the formulas, B2 refers to the green band, and B4 refers to the near-infrared band.

Next, based on the preliminary classification results of the image, target areas covering the 6 land cover classes are selected within the image. The visual interpretation method is employed to delineate the land cover classes in the target areas as polygon features and assign corresponding attribute values. In this case, 1 represents cropland, 2 represents forest land, 3 represents grassland, 4 represents water bodies, 5 represents urban, industrial, and residential land, and 6 represents unused land. Finally, the interpreted results are corrected through expert judgment and field investigation, and the vector data are converted into raster data, serving as the ground truth label dataset. The processed image data and their corresponding label data are divided into training, testing, and validation sets using a sliding window approach in an 8:1:1 ratio. The final dataset consists of 1369 images in the training set and 329 and 327 images in the testing and validation sets, respectively. Some sample images and their corresponding labels are shown in Figure 3.

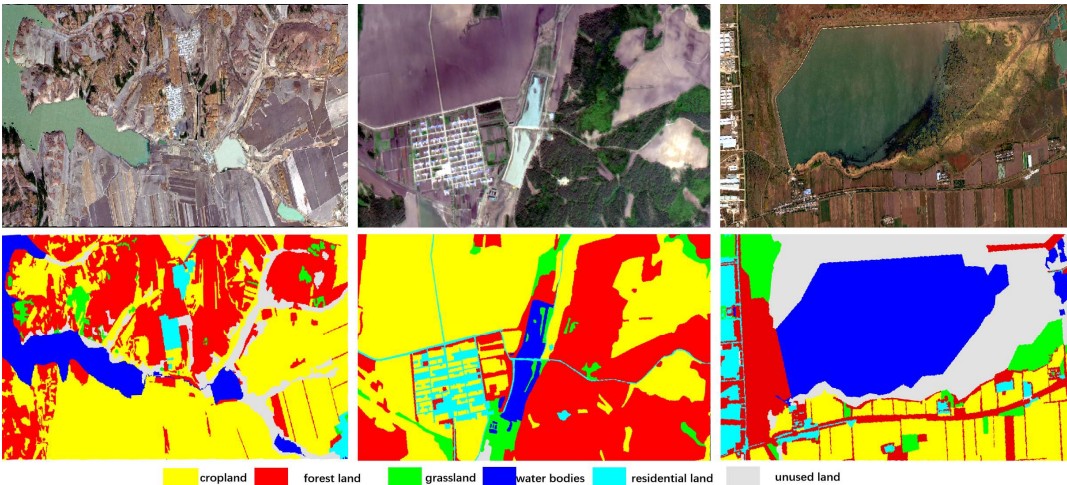

| cropland | forest land | grassland | water bodies | residential land | unused land |

**Figure 3.** Dataset images and corresponding labels.

### 2.3. Deep Learning Method

In this study, a personally constructed sample library was used as the data source. The U-Net model was chosen as the backbone network, and a contracting path with residual modules was employed to extract contextual features of the land objects, aiming to improve feature extraction accuracy and avoid the degradation of the model. The size of the convolutional kernels in the model was adjusted to explore the impact of different kernel sizes on the accuracy of land feature extraction. Finally, the ResNet34 residual module and $3 \times 3$ convolution were selected to construct the RAT-UNet model.

As shown in Figure 4, the input is a $256 \times 256$ image. Firstly, a $3 \times 3$ convolution is applied once to adjust the number of input channels, facilitating the subsequent residual connection calculations. Then, four residual connection calculations are performed. After each residual connection calculation, max pooling with a stride of 2 is used for downsampling. Subsequently, the image enters the expansion path of the model, where each layer employs a $2 \times 2$ transpose convolution for upsampling. Simultaneously, skip connections are used to fuse shallow and deep feature maps. Then, two $3 \times 3$ convolutions are applied. After four iterations of upsampling, convolution, and feature fusion, semantic segmentation of the image is achieved. Finally, a $3 \times 3$ convolution is performed to adjust the channel number to the number of extracted classes, resulting in a $256 \times 256$ output image representing the extracted land cover classes in the black soil region. The structure of the RAT-UNet model is illustrated below.

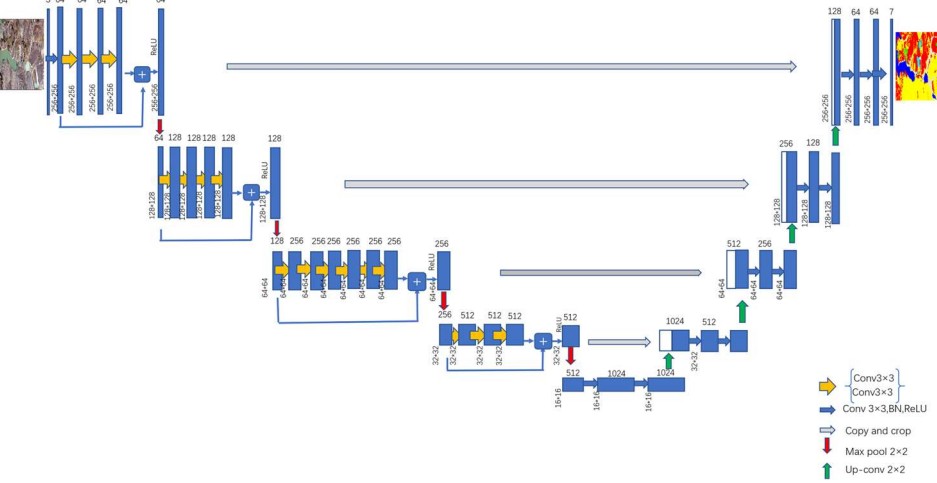

**Figure 4.** RAT-UNet architecture.

*2.4. Accuracy Assessment Method*

The accuracy evaluation method is used to assess the performance of the model. After the model training is completed, the model's extraction results need to be evaluated for accuracy. Common accuracy evaluation metrics in deep learning include Overall Accuracy (OA), Precision (P), Recall (R), F1 score, Intersection over Union (IoU), and Mean Intersection over Union (MIoU). OA represents the percentage of correctly predicted samples among all samples. A higher OA indicates that the model has predicted more samples correctly overall. F1 score is a comprehensive metric that is the harmonic mean of Precision and Recall. It provides a balanced measure of the model's performance in both Precision and Recall. IoU measures the overlap between the target region and the predicted region, with values ranging from 0 to 1. A higher IoU indicates better prediction performance. IoU is calculated for each individual class, while MIoU is the average IoU across all classes. Therefore, in this study, OA, F1 score, and MIoU are selected as the evaluation metrics to assess the model's land cover extraction performance. The formulas for calculating OA, F1 score, and IoU are shown in Equations (3)–(7):

$$\text{Accuracy} = \frac{\text{TP} + \text{TN}}{\text{TP} + \text{TN} + \text{FP} + \text{FN}} \tag{3}$$

$$\text{F1 score} = 2 \times \frac{\text{Precision} \times \text{Recall}}{\text{Precision} + \text{Recall}} \tag{4}$$

$$\text{IoU} = \frac{\text{TP}}{\text{TP} + \text{FP} + \text{FN}} \tag{5}$$

$$\text{Precision} = \frac{\text{TP}}{\text{TP} + \text{FP}} \tag{6}$$

$$\text{Recall} = \frac{\text{TP}}{\text{TP} + \text{FN}} \tag{7}$$

In the equations: TP represents the number of pixels correctly classified as positive samples in the classification result; TN represents the number of pixels correctly classified as negative samples in the classification result; FP represents the number of pixels incorrectly classified as positive samples in the classification result; FN represents the number of pixels incorrectly classified as negative samples in the classification result; Precision represents the accuracy of the positive predictions, calculated as TP divided by the sum of TP and FP; Recall represents the rate of correctly identified positive samples, calculated as TP divided by the sum of TP and FN.

## 3. Experimental Setup and Results Analysis

*3.1. Experimental Setup*

The experiments in this study were conducted using the PyTorch deep learning framework. To ensure fairness and consistency, all model algorithms were trained using the Adaptive Moment Estimation (Adam) optimizer with a $2 \times 10^{-4}$ learning rate. The batch size was set to 2 samples, and the models were trained for 200 epochs. During the model training process, we closely monitored the changes in validation accuracy. The parameters of the model from the last 10 training iterations were saved, and the model with the highest validation accuracy was selected for testing purposes. This approach ensured that the best-performing model was used for evaluating the results.

The RAT-UNet model was utilized for land cover extraction in the black soil region, and its effectiveness was validated through a series of comparative experiments. Firstly, a comparative experiment was conducted between the RAT-UNet model and traditional models such as U-Net, SegNet, LinkNet34, and DeepLabV3. The aim was to analyze the advantages and disadvantages of the RAT-UNet model. Secondly, comparative experiments were performed with different numbers of residual network layers and different kernel sizes

to validate the effectiveness of the RAT-UNet model. Next, comparative experiments were conducted using different band combinations to determine the optimal band combination for land cover extraction in the black soil region. Finally, ablation experiments were carried out to analyze the roles played by the Resnet34 residual module and the 3 × 3 convolution in land cover extraction in the black soil region. Through these experiments, the performance and effectiveness of the RAT-UNet model were evaluated, providing insights into its strengths and optimal configurations for land cover extraction in the black soil region.

### *3.2. Results Analysis*

3.2.1. Large-Scale Area Land Cover Extraction Results

Utilizing the RAT-UNet model to extract land cover information in Qiqihar City, a representative black soil region of Northeast China, can provide valuable data support for the monitoring and management of regional natural resources. As shown in Figure 5, the RAT-UNet model can accurately extract most of the land coverage information, among which the extraction performance of cropland, grassland, water bodies and residential land is good, which can provide effective data for large-scale land cover monitoring in the black soil region. This, in turn, can facilitate the formulation and implementation of land planning and management decisions in the black soil region. However, for scattered grassland and unused land, the RAT-UNet model shows noticeable misclassifications and omissions, as indicated by the rectangles in the figures. The confusion between unused land and grassland, as well as the misclassification of unused land as cropland or other categories, may be attributed to data imbalance. In future research, further improvements can be made to the model to address issues such as data imbalance.

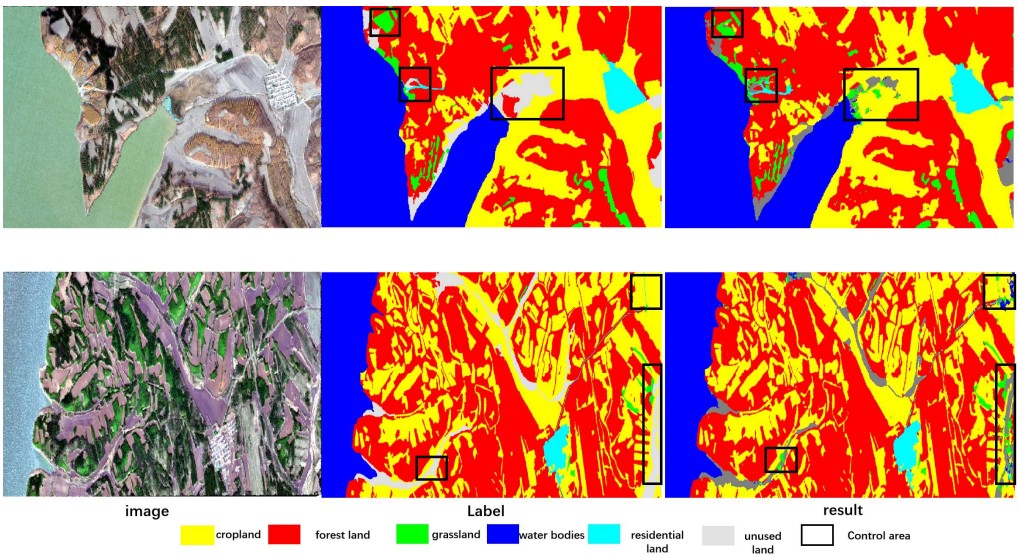

**Figure 5.** Large-Scale Area Land Cover Extraction Results.

3.2.2. Local Area Land Cover Extraction Results

By conducting comparative experiments between the RAT-UNet model and U-Net, SegNet, LinkNet34, and DeepLabV3 models, the land cover extraction results from these five models are thoroughly analyzed. This comparative analysis aims to validate the effectiveness of the RAT-UNet model and affirm its potential as a reliable and efficient approach for land cover extraction in the study area.

As shown in Table 1, the RAT-UNet model achieved the highest overall accuracy of 93.04% in land cover extraction, surpassing the performances of DeepLabV3, U-Net, SegNet, and LinkNet34 models in sequence. The RAT-UNet model demonstrated superior accuracy in extracting land cover information, thus providing effective data support for land

planning and management in the black soil region. Regarding the extraction performance of the six land cover categories, the RAT-UNet model achieved the highest extraction accuracy for cultivated land, followed by water bodies, forest land, residential land, unused land, and grassland. Except for grassland, the extraction accuracy for other land cover categories exceeded 80%. This high level of accuracy can provide effective methodological support for the monitoring and management of cultivated land resources in the black soil region. The timely acquisition of cultivated land cover information can facilitate the implementation of policies for the development and conservation of cultivated land resources in the area. Compared to the four classical deep learning models, the RAT-UNet model demonstrates superior extraction performance for cultivated land, forest land, and residential land. Although the extraction performance for water bodies, grassland, and unused land is slightly lower than that of the DeepLabV3 model, it still provides valuable insights for policy-making in the black soil region, especially for policies related to afforestation, farmland conservation, and urban–rural planning.

**Table 1.** Land cover extraction accuracy evaluation table for five models.

| F1score | RAT-UNet | DeepLabV3 | U-Net | SegNet | LinkNet34 |
|---|---|---|---|---|---|
| cropland | 95.11% | 94.10% | 93.47% | 86.09% | 83.62% |
| forest land | 93.61% | 93.30% | 91.47% | 83.71% | 81.18% |
| grassland | 68.41% | 68.62% | 63.97% | 53.83% | 32.19% |
| water bodies | 94.67% | 95.20% | 92.34% | 83.02% | 71.89% |
| residential land | 89.40% | 88.79% | 87.84% | 80.94% | 75.07% |
| unused land | 87.25% | 88.47% | 81.40% | 69.03% | 52.80% |
| OA | 93.04% | 92.81% | 90.77% | 82.67% | 77.31% |
| MIoU | 79.79% | 79.75% | 75.29% | 62.67% | 51.91% |

To further analyze the advantages of the RAT-UNet model, a visual analysis of land cover extraction results in specific local regions can be conducted. As shown in Figure 6A–F, image A–F corresponds to cropland, forest, grassland, water bodies, residential land, and unused land, respectively. The rectangular boxes in each image highlight the differences in land cover extraction among the five models. The black rectangular boxes represent the ground truth land cover labels, the blue rectangular boxes represent correctly extracted areas, and the orange rectangular boxes represent incorrectly extracted areas. Visual analysis of the extraction results from the five models reveals the following observations: The RAT-UNet model demonstrates an advantage in extracting linear features and is capable of preserving complete boundary information of land cover. The DeepLabV3 model can extract more complete information about water bodies, but its performance in extracting small and scattered features is relatively poor. The U-Net model lacks the ability to extract small linear features, and the extracted land cover boundaries appear slightly blurred. The SegNet model can only extract a limited number of linear features, and the extracted land cover boundaries are relatively unclear. The LinkNet34 model performs well in extracting cropland and residential land, but it shows evident misclassification issues when dealing with areas where multiple land cover types are mixed.

In conclusion, for the purpose of monitoring and managing the black soil resources in Qiqihar City, the RAT-UNet model outperforms the DeepLabV3, U-Net, SegNet, and LinkNet34 models in land cover extraction, with an overall accuracy of 93.04%. It shows particularly high accuracy in extracting cropland, which is beneficial for subsequent cropland resource planning and management tasks. The RAT-UNet model demonstrates certain advantages in extracting linear features and retains complete boundary information of land objects. It provides accurate and comprehensive land cover information for regional natural resource monitoring and management. However, the RAT-UNet model still exhibits limitations in dealing with irregularly distributed small features, as shown in Figure 6E, where there is a missing extraction of cropland around the periphery of residential land, and

in Figure 6D, where water bodies are misclassified as unused land. Further improvements can be made to the model to address these issues in future research.

**Figure 6.** (**A–F**) Land cover extraction results of five models.

## 4. Discussion

### 4.1. Impact of Residual Network Depth on Model Extraction Accuracy

The contraction path of the RAT-UNet model utilizes residual modules for contextual feature extraction. To investigate the impact of residual network depth on the extraction accuracy of the model, we conducted experiments using both Resnet18 and Resnet34 residual modules. Figure 7 illustrates the accuracy and loss curves of the land cover extraction models using these two residual modules. The x-axis represents the epochs, while the y-axis represents the loss and accuracy, respectively. After training for 150 epochs, the accuracy and loss curves of the models using both residual modules were comparable. However, the curve using the Resnet34 residual module exhibited less fluctuation, indicating that the land cover extraction model with Resnet34 converged faster. Therefore, the RAT-UNet model selected the Resnet34 residual module for contextual feature extraction in the contraction path.

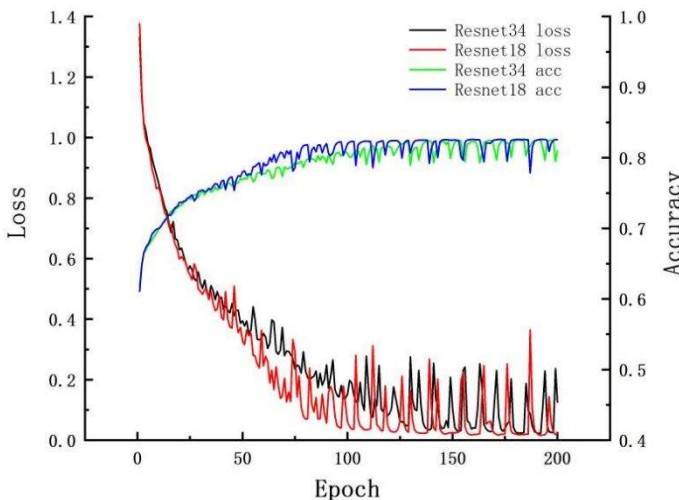

**Figure 7.** Accuracy and loss curves of different residual modules.

To further analyze the effectiveness of the Resnet34 residual module, we compared the land cover extraction performance of models using both Resnet18 and Resnet34 residual modules for the six land cover classes. In the bar chart shown in Figure 8, the x-axis represents the six land cover classes, and the y-axis represents the F1 score of land cover extraction. Observing the bar chart, we can see that both models using the two residual modules achieved extraction accuracy above 90% for cropland, forest, and water bodies. The extraction accuracy for residential land and unused land was above 80% for both models. The model using the Resnet34 residual module showed improvements of 0.91%, 1.29%, 12.11%, 3.37%, 0.93%, and 5.82% in extraction accuracy for the six land cover classes compared to the model using Resnet18. Therefore, the RAT-UNet model selected the Resnet34 residual module for the contraction path as it demonstrated higher land cover extraction performance.

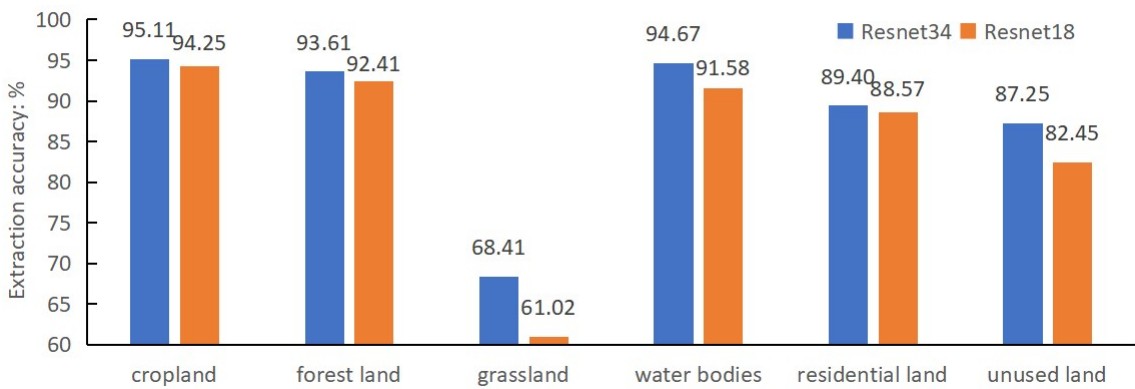

**Figure 8.** Comparison plot of ground object extraction accuracy for different residue modules.

### 4.2. The Impact of Convolutional Kernel Size on Model Extraction Accuracy

The first convolutional layer of the RAT-UNet model is used to adjust the number of channels, converting the feature channels from 3 to 64. This adjustment facilitates the subsequent residual connections. The convolutional kernel can be seen as a filtering operator in digital image processing, which performs weighted summation over local regions of the input feature map to extract features. Commonly used convolutional kernel sizes include $1 \times 1$, $3 \times 3$, $5 \times 5$, and $7 \times 7$. In the Resnet34 network, the first convolutional layer is a $7 \times 7$ convolution with a stride of 2. To investigate the impact of convolutional kernel size on feature extraction, experiments were conducted using $7 \times 7$, $5 \times 5$, $3 \times 3$, and $1 \times 1$ convolutions for the first layer. As shown in Table 2, the model with a $3 \times 3$ kernel in

the first convolutional layer achieved the highest extraction accuracy. This result indicates that a larger kernel size does not necessarily lead to better feature extraction. Larger kernels increase the complexity of the convolutional computation and involve more parameters in the calculation. On the other hand, smaller kernels result in simpler computations with fewer parameters, but the extracted features may not be as clear. Therefore, the selection of kernel size should not be too large or too small. The RAT-UNet model sets the kernel size of the first convolutional layer in the residual module to 3 × 3, reducing the number of parameters while maintaining good feature extraction performance.

**Table 2.** Land cover extraction accuracy table of different convolutions in the first layer.

| Convolutional Kernel Size | Metric Category | Cropland | Forest Land | Grassland | Water Bodies | Residential Land | Unused Land |
|---|---|---|---|---|---|---|---|
| 1 × 1 | F1score | 89.17% | 85.38% | 40.27% | 85.44% | 85.00% | 68.76% |
| | IoU | 80.45% | 74.48% | 25.21% | 74.58% | 73.91% | 52.39% |
| | OA | 84.52% | | | | | |
| 3 × 3 | F1score | 95.11% | 93.61% | 68.41% | 94.67% | 89.40% | 87.25% |
| | IoU | 90.67% | 87.98% | 51.98% | 89.88% | 80.84% | 77.38% |
| | OA | 93.04% | | | | | |
| 5 × 5 | F1score | 88.19% | 84.03% | 25.61% | 84.50% | 78.66% | 60.89% |
| | IoU | 78.87% | 72.46% | 14.68% | 73.16% | 64.82% | 43.77% |
| | OA | 82.44% | | | | | |
| 7 × 7 | F1score | 93.98% | 91.16% | 66.18% | 94.05% | 87.94% | 83.74% |
| | IoU | 88.65% | 83.76% | 49.45% | 88.76% | 78.47% | 72.03% |
| | OA | 91.33% | | | | | |

The last layer of the RAT-UNet model, similar to U-Net, is used to adjust the number of channels to match the number of output classes for convenient feature output. In U-Net, the last layer consists of a 1 × 1 convolution. In this study, we conducted experiments using both 1 × 1 and 3 × 3 convolutions as the last layer of the model to investigate the impact of the convolution kernel size on the model's extraction accuracy.

Observing the extraction results in Table 3, it can be seen that compared to the model with a 1 × 1 convolution as the last layer, the model with a 3 × 3 convolution as the last layer shows an improvement in Intersection over Union (IoU) for the categories of cropland, forest land, grassland, residential land, and unused land, with increases of 0.5%, 1.83%, 0.66%, 2.04%, and 2.06%, respectively. However, the IoU for water bodies decreased by 0.54%. Overall, considering the improvement in IoU for the other five land categories and the overall accuracy increase of 0.43%, setting the last layer's convolution kernel size to 3 × 3 demonstrates better land extraction performance.

**Table 3.** Land cover extraction accuracy table of different convolutions in the last layer.

| Kernel Size | Accuracy Categories | Cropland | Forest Land | Grassland | Water Bodies | Residential Land | Unused Land |
|---|---|---|---|---|---|---|---|
| 1 × 1 | F1score | 94.86% | 92.69% | 68.11% | 94.94% | 88.38% | 86.23% |
| | IoU | 90.22% | 86.37% | 51.64% | 90.37% | 79.19% | 75.79% |
| | OA | 92.64% | | | | | |
| 3 × 3 | F1score | 95.11% | 93.61% | 68.41% | 94.67% | 89.40% | 87.25% |
| | IoU | 90.67% | 87.98% | 51.98% | 89.88% | 80.84% | 77.38% |
| | OA | 93.04% | | | | | |

### 4.3. Impact of Band Combination Methods on Model Accuracy

Different band combination methods in remote sensing imagery can have an impact on the accuracy of land cover extraction. The reflectance of black soil in the study area is significantly influenced by soil organic matter content. Experimental results of soil organic matter retrieval using high spatial resolution remote sensing data indicate that the highest

correlation exists between black soil organic matter content and the infrared and near-infrared bands. The band combination of green, red, and near-infrared (b2, b3, b4) yields the best results for extracting black soil information. In order to explore band combination methods more suitable for land cover extraction in black soil regions, four different band combinations were utilized: blue, green, red (b1, b2, b3); green, red, near-infrared (b2, b3, b4); blue, green, near-infrared (b1, b2, b4); and blue, red, near-infrared (b1, b3, b4). The results of land cover extraction using these band combinations are shown in Figure 9. It can be observed that the model performance using the band combination of b1, b2, b3 achieves the best results, particularly for grassland and unused land, with significant improvements in accuracy compared to the other three band combinations (b2, b3, b4; b1, b2, b4; and b1, b3, b4). The accuracy of grassland extraction increased by 52.43%, 40.71%, and 49.67%, respectively, while the accuracy of unused land extraction increased by 34.43%, 23.86%, and 38.9%, respectively. Therefore, this study suggests that the best band combination for land cover extraction in black soil regions is based on the blue, green, and red bands.

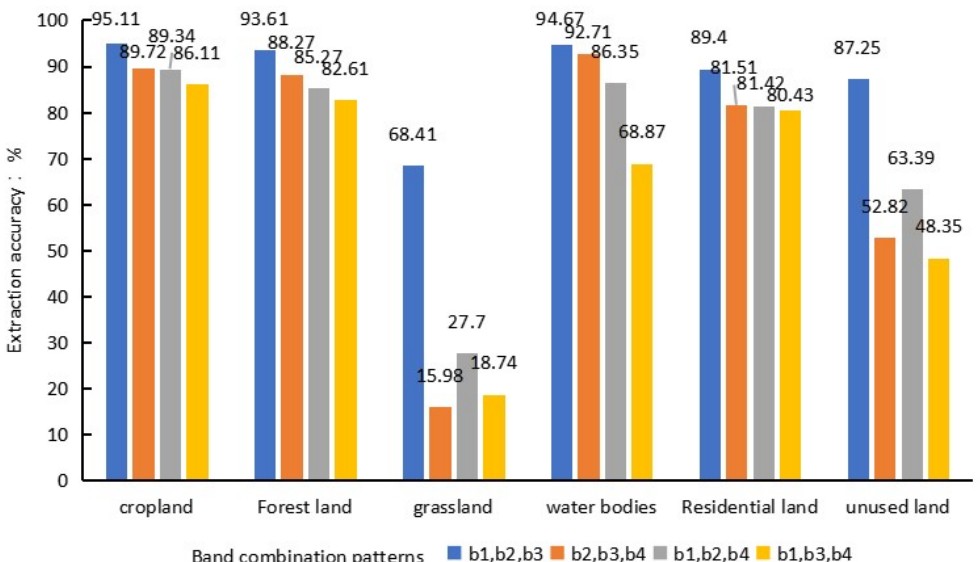

**Figure 9.** Comparison plot of land cover extraction accuracy for different band combination patterns.

*4.4. Ablation Experiment*

To verify the effectiveness of the residual module and the 3 × 3 convolution in land cover extraction, two ablation experiments were conducted in this study: ① Restoring the final 3 × 3 convolution to a 1 × 1 convolution (RAT-UNet1×1). ② Restoring the residual modules in the contracting path to regular convolutional modules (U-Net). The extraction results are shown in Table 4 and Figure 10A–C.

**Table 4.** Evaluation table of ablation experiments.

| Model | Final Convolution 3 × 3 | Residual Module | Final Convolution 1 × 1 | MIoU | OA |
|---|---|---|---|---|---|
| RAT-UNet | √ | √ | | 79.79% | 93.04% |
| RAT-UNet1×1 | | √ | √ | 78.93% | 92.64% |
| U-Net | | | √ | 69.84% | 88.23% |

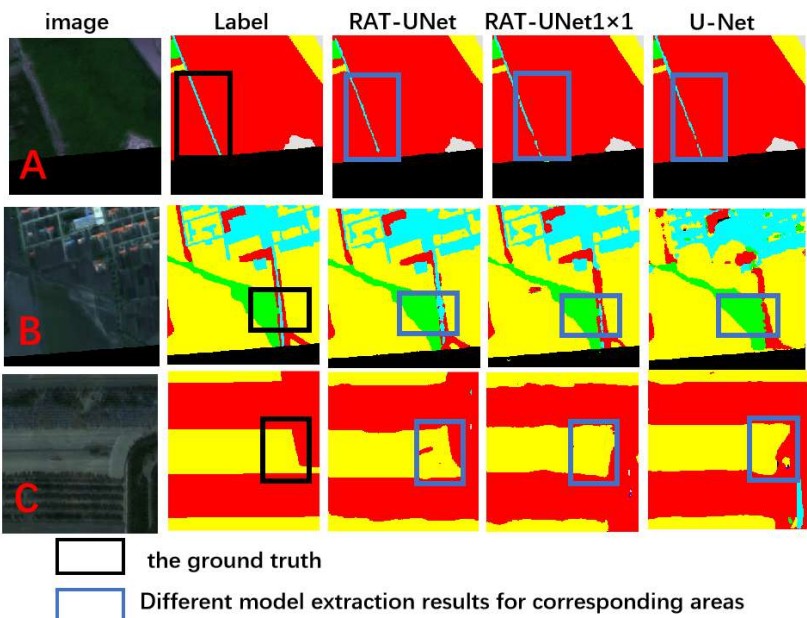

**Figure 10.** (**A**–**C**) The object extraction results of ablation experiments.

The two ablation experiments verified the effectiveness of the RAT-UNet model. The RAT-UNet1×1 model, which has a 1 × 1 convolutional kernel in the final layer, showed a decrease of 0.43% in overall extraction accuracy. It exhibited difficulties in extracting mixed land cover classes and lacked comprehensive feature information compared to the RAT-UNet model. For example, in Figure 10B, there is confusion between forest, grassland, and residential areas in the rectangular area. This result indicates that the 3 × 3 convolutional kernel can learn more discriminative features, enabling more effective differentiation of various land cover classes.

On the other hand, the U-Net model with regular convolutional modules replacing the residual modules in the contracting path showed a decrease of 5.45% in overall extraction accuracy compared to the RAT-UNet model. It failed to extract small linear features and exhibited significant misclassification of land cover. For instance, in Figure 10B, the rectangular area shows a lack of linear features in the land cover, and there is confusion between residential land and cropland. In Figure 10C, forest land was misclassified as residential land. This result suggests that using residual modules in the contracting path of the model is advantageous for the extraction of small linear features.

Based on the comprehensive analysis and comparison of the results from the ablation experiments mentioned above, it can be concluded that the residual modules in the RAT-UNet model are beneficial for the extraction of small linear features. Additionally, using a 3 × 3 convolutional kernel helps in learning more informative features.

## 5. Conclusions

As agricultural development and urbanization continue to progress, the issue of land use in the black soil region of Northeast China has become severe. Currently, obtaining land cover information in the black soil region relies on traditional methods, which are difficult to meet the demands for accurate and timely monitoring. In this study, we constructed the RAT-UNet land cover extraction model based on deep learning methods. Taking Qiqihar City as an example, we used the RAT-UNet model for land cover extraction, achieving an impressive overall accuracy of 93.04%. Through a series of comparative experiments, we demonstrated that RAT-UNet outperforms traditional models such as DeepLabV3, U-Net, and SegNet in land cover extraction performance. Additionally, we conducted ablation experiments that showcased the advantages of the residual network in extracting linear land features.



Indeed, the RAT-UNet model still exhibits misclassification and omission issues when it comes to extracting irregularly distributed small land features and land cover classes with limited data samples. In future research, we can explore ways to improve the performance of land cover extraction models by focusing on boundary loss functions and data augmentation techniques. These approaches may help enhance the model's ability to accurately identify and extract challenging land cover categories and improve its overall performance.

**Author Contributions:** Writing—original draft, B.D. and T.Z.; Supervision, J.T. and Y.W. All authors have read and agreed to the published version of the manuscript.

**Funding:** This research was partly supported by the Strategic Priority Research Program of Chinese Academy of Sciences (Grant No. XDA28050200) and Hunan Provincial Postgraduate Scientific Research and Innovation Project (Grant No. CX20211059).

**Data Availability Statement:** Due to the nature of this research, participants of this study did not agree for their data to be shared publicly, so supporting data is not available.

**Conflicts of Interest:** The authors declare no conflict of interest.

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
