# Peer review of "Land Cover Extraction in the Typical Black Soil Region of Northeast China Using High-Resolution Remote Sensing Imagery"

_land, doi:10.3390/land12081566_

Round 1
Reviewer 1 Report
The authors attempt to introduce residual modules and adjust the convolutional kernel size to build a land cover extraction model suitable for the black soil region and to improve the accuracy and efficiency of obtaining land cover information and provide timely data for regional natural resource planning and management in Qiqihar City. It is very interesting. The strength of this paper is that the authors tried to develop a high-precision land feature extraction model called RAT-UNet, which achieves high accuracy in land feature extraction. Amount of data and work have been carried out. However, the English needs to be perfected, as there are still quite a few grammatical errors in the current manuscript. Moreover, there are some questions lacking deep thoughts and the authors do not work them out. Overall, the manuscript is easy to follow but great efforts should be taken before the manuscript could be published. A minor revision in needed. The following comments could be used as a guideline when the authors think about how to revise this manuscript.
1. Introduction:
The organization and language of the introduction should be improved with great efforts. There are many problems with sentence structure and clause, which lead to misunderstanding by readers. The ground object extraction method based on traditional method and the ground object extraction method based on new learning are respectively described. The classification of the two methods is unclear. Please reorganize those sentences.
2. Methods:
This part is a little long. It is needed more patient for the reviewer to read the section. Please highlight the important and unusual methods. The methods often used such as NDVI and NDWI, can be introduced shortly. Moreover, in the Section 2.4, the accuracy evaluation indicators deal with OA, F1 score, IoU and other indicators. Why choose these three indicators? What is the role of each indicator?
3. Results and analysis:
Overall accuracy and average crossover ratio are used as reference indicators to evaluate the accuracy of ground objects extraction. Why is the average crossover ratio used here in section 3.2 and what is the meaning of this index? What is the difference with the intersection ratio in the accuracy evaluation method above? It should be elaborated clearly.
4. Conclusion:
The conclusion of this manuscript should highlight the innovation of this study and the uniqueness and validity of the RAT-UNet model.
5. Figures and Tables:
>Chinese map in Figure 1 is not standardized. Revised Figure 1 according to the Ministry of Natural Resources of the People’s Republic of China. Moreover, please add names to the highlighted areas in the three maps so that readers can understand the study area better.
>The pictures in Figure 3 are not representative and can be changed to example maps of images and labels in a large area of the study area.
>Try to improve the resolution of Figure 4.
> Figure 5.D and Figure 5.E are not representative and fail to reflect the advantages of the RAT-UNet model.
> The description of Table 1 needs to be reorganized, focusing on the advantages and uniqueness of the RAT-UNet model compared with the other four models, as well as its important contribution to the black soil area.
>The titles of the tables and the figures should be complete.
The manuscript is well written and easy to follow. However, the English needs to be perfected, as there are still a few grammatical errors in the current manuscript.Author Response
Response to Reviewer 1 Comments
Point 1:The organization and language of the introduction should be improved with great efforts. There are many problems with sentence structure and clause, which lead to misunderstanding by readers. The ground object extraction method based on traditional method and the ground object extraction method based on new learning are respectively described. The classification of the two methods is unclear. Please reorganize those sentences.
Response 1: We have reorganized and improved the introduction, carefully checked the grammar expressions in the article, and further refined the English. Regarding the classification of traditional and new methods for land cover extraction, a clearer categorization has been established. The traditional land cover extraction methods include field survey methods and remote sensing-based extraction methods, while the new methods consist of AI-based extraction methods, which encompass machine learning and deep learning approaches.
Point 2: Methods part is a little long. It is needed more patient for the reviewer to read the section. Please highlight the important and unusual methods. The methods often used such as NDVI and NDWI, can be introduced shortly. Moreover, in the Section 2.4, the accuracy evaluation indicators deal with OA, F1 score, IoU and other indicators. Why choose these three indicators? What is the role of each indicator?
Response 2: In the Materials and Methods section, we introduced the data sources, construction of the land cover sample library, deep learning method, and accuracy evaluation method. Regarding the land cover construction part, we mentioned the preliminary extraction of vegetation and water bodies based on remote sensing principles using NDVI and NDWI. Since these two remote sensing indices are commonly used, providing an in-depth explanation of their principles may not be necessary. Therefore, we have removed the detailed explanation and only briefly mentioned the calculation formulas for NDVI and NDWI.As for the accuracy evaluation method,we only mentioned that OA, F1 score, and IoU were selected as the evaluation metrics without providing a rationale for their selection. To address this,we have supplemented the section with common evaluation metrics used in deep learning and explained the role and significance of OA, F1 score, and IoU in assessing the model's performance. This will provide a more informed basis for the selection of these evaluation metrics.
Point 3:Results and analysis part, overall accuracy and average crossover ratio are used as reference indicators to evaluate the accuracy of ground objects extraction. Why is the average crossover ratio used here in section 3.2 and what is the meaning of this index? What is the difference with the intersection ratio in the accuracy evaluation method above? It should be elaborated clearly.
Response 3:The model's accuracy evaluation metrics include Overall Accuracy and Mean Intersection over Union (MIoU), which can provide a comprehensive assessment of the model's performance globally. IoU and MIoU both represent the overlap between the target region and the predicted region, where IoU calculates the result for an individual class, while MIoU is the average Intersection over Union across all classes. MIoU is widely used in both natural image and remote-sensing image semantic segmentation. The direct use of MIoU in the Results and Analysis section may seem abrupt, so we have supplemented relevant content about MIoU in the accuracy evaluation method section to ensure a more logical flow of the article.
Point 4:The conclusion of this manuscript should highlight the innovation of this study and the uniqueness and validity of the RAT-UNet model.
Response 4:We have revised the conclusions to emphasize the novelty of our study in addressing the limitations of traditional land cover extraction methods in the black soil region. We proposed the RAT-UNet model based on deep learning methods, highlighting its innovative approach. By focusing on the extraction accuracy of the RAT-UNet model, we demonstrated its effectiveness and uniqueness. Through a comprehensive analysis of the experimental results, we identified the limitations of the RAT-UNet model and provided corresponding prospects for future research, laying the foundation for further advancements in this field.
Point 5:Figures and Tables:â‘ Chinese map in Figure 1 is not standardized. Revised Figure 1 according to the Ministry of Natural Resources of the People’s Republic of China. Moreover, please add names to the highlighted areas in the three maps so that readers can understand the study area better.â‘¡The pictures in Figure 3 are not representative and can be changed to example maps of images and labels in a large area of the study area.â‘¢Try to improve the resolution of Figure 4.â‘£Figure 5.D and Figure 5.E are not representative and fail to reflect the advantages of the RAT-UNet model.⑤The description of Table 1 needs to be reorganized, focusing on the advantages and uniqueness of the RAT-UNet model compared with the other four models, as well as its important contribution to the black soil area.â‘¥The titles of the tables and the figures should be complete.
Response 5:We have made modifications to the figures and tables in the article. We followed the standardization of Figure 1, which depicts the map of China based on the revision of the Ministry of Natural Resources of the People's Republic of China, and added names to highlight specific areas. In Figure 3, we revised the label map and image to select more representative regions. We also enhanced the resolution of Figure 4 to 300 dpi. For Figures 5.D and 5.E, we focused on water bodies and residential land, respectively, to visually analyze the extraction results, reflecting the effectiveness of the RAT-UNet model. Additionally, we updated the description in Table 1 to present the experimental results of the RAT-UNet model's overall accuracy, land cover class extraction accuracy, and comparisons with five other models. The revised table emphasizes the effectiveness and uniqueness of the RAT-UNet model, as well as its contribution to the black soil region. Furthermore, we improved the titles of the figures and tables to enhance clarity and understanding.
Reviewer 2 Report
Overall, the study is important for land cover extraction in black soil areas, but there is room for improvement in the writing and presentation of the article. I hope you will consider and make adjustments based on the suggested revisions to improve the completeness and persuasiveness of this paper. I recommend accepting the paper for publication with these minor revisions.
1. The paper makes extensive use of abbreviations and jargon, which can make it difficult for readers, especially those less familiar with the field, to understand. I recommend avoiding excessive use of abbreviations and jargon, especially when they first appear. Instead, provide the full name or definition in parentheses immediately after introducing the abbreviation or term. This will ensure that readers can easily grasp the meaning and context of the terms used in your research. For example, RAT-UNet in Section 2.3.
2. In the Results Analysis section, I suggest providing more quantitative analysis and statistical information for comparing the results of the experiments. The authors could also consider using other evaluation metrics and visualisation methods to better demonstrate the performance of the models.
3. In the Discussion section, you can further discuss the limitations of the model and potential directions. For example, optimization strategies and methods to enhance the model are suggested for the omission extraction and misclassification problems, or discuss how future data and technological advances may address these problems.
4. In the "Conclusion" section, authors can summarise more clearly the main contributions and innovations of this research and provide an outlook and suggestions for future work. This will help readers to better understand the results of this research and provide direction and insights for further research in related areas.
Author Response
Response to Reviewer2 Comments
Point 1:The paper makes extensive use of abbreviations and jargon, which can make it difficult for readers, especially those less familiar with the field, to understand. I recommend avoiding excessive use of abbreviations and jargon, especially when they first appear. Instead, provide the full name or definition in parentheses immediately after introducing the abbreviation or term. This will ensure that readers can easily grasp the meaning and context of the terms used in your research. For example, RAT-UNet in Section 2.3.
Response 1:The abbreviations in the article are all written in full when first mentioned, followed by the abbreviation in parentheses. For example, ConvolutionalNeuralNetwork (CNN), and thereafter, ConvolutionalNeuralNetwork is represented as CNN. The RAT-UNet model mentioned in section 2.3 is a nomenclature proposed by the authors for the land cover extraction model in this paper, and it is not an abbreviation.
Point 2: In the Results Analysis section, I suggest providing more quantitative analysis and statistical information for comparing the results of the experiments. The authors could also consider using other evaluation metrics and visualisation methods to better demonstrate the performance of the models.
Response 2: The Results and Analysis section provides detailed explanations and discussions from two perspectives: the land cover extraction results for the large-scale area and the land cover extraction results for the local area. This enriched the original experimental findings, enabling a better demonstration of the performance of the RAT-UNet model.
Point 3: In the Discussion section, you can further discuss the limitations of the model and potential directions. For example, optimization strategies and methods to enhance the model are suggested for the omission extraction and misclassification problems, or discuss how future data and technological advances may address these problems.
Response 3:The discussion section of the paper focuses on a series of comparative experiments, including the impact of residual network layers, convolutional kernel sizes, and band combinations on the model's extraction performance. Additionally, it demonstrates the effectiveness of newly added network modules through ablation experiments. While the results section already mentioned the limitations of the RAT-UNet model, the discussion section does not further elaborate on its limitations and potential directions for improvement.
Point 4:In the "Conclusion" section, authors can summarise more clearly the main contributions and innovations of this research and provide an outlook and suggestions for future work. This will help readers to better understand the results of this research and provide direction and insights for further research in related areas.
Response 4:We have revised the conclusion to emphasize the novelty of this research. The conclusion section now starts by highlighting the current land use issues in the black soil region of Northeast China. It then introduces the novel RAT-UNet model proposed in this study and presents a series of experimental results to underscore its effectiveness and innovation. Finally, the conclusion provides insights and directions for further research, addressing the existing issues in the model and offering guidance for future studies in related fields.
Reviewer 3 Report
This study compared different deep learning algorithms for land cover type extraction and concluded the superiority of the proposed deep learning model RAT-UNet. The manuscript is generally well drafted and can be easily understood. However, I have several major comments listed as below.
Authors used “Land feature extraction” in the title, abstract and many places in the Introduction, which means extraction of six land cover types. However, I would consistently convert it to “land cover extraction” or “land cover feature extraction” through the paper. The meaning of “land feature” is not clear unless authors clarify it.
Lines 263-273 clearly state the four objectives of this study, which correspond to Results 3.2, Dicussion4.1, Dicussion.4.2, and Dicussion.4.3, respectively. I would incorporate 4.1-4.3 into Results because they are parts of the results, not extra discussions. Authors may combine all these subtitles into one “Results and Discussion” section.
Figure 3 is not clear, less than 300dpi.
Figure 5, Please directly add the names of each land cover type for A-F, and also add a legend for the colors.
Figure 7. What does y axis represent, add the y axis label.
Table 2 and 3. Some land cover names have capitalized initial letter, some don’t have.
Section 4.3 and Figure 8. I think the design of these four data combinations doesn’t make any sense. I am more interested in a data combination design for exploring the contribution of visible bands, NIR band, and vegetation indices in land cover type extraction using deep learn model, like b1+b2+b3, b1+b2+b3+b4, and b1+b2+b3+ ndvi+ndwi, and b1+b2+b3+b4+ ndvi+ndwi. b1-b3 represent the basic visible optical bands, the NIR band (b4) and two vegetation indices represent extra or enhanced data information. Also as authors mentioned in Lines 184-200, NIR actually plays important roles in identifying vegetation and water bodies, why the data combination without band 4 (i.e., band 1+band2+band3) obtained highest classification accuracy in vegetation and water body identification (Figure 8). Authors may need to discuss it in the manuscript.
Author Response
Response to Reviewer3 Comments
Point 1:Authors used “Land feature extraction” in the title, abstract and many places in the Introduction, which means extraction of six land cover types. However, I would consistently convert it to “land cover extraction” or “land cover feature extraction” through the paper. The meaning of “land feature” is not clear unless authors clarify it.
Response 1:The RAT-UNet model constructed in this study is used for land cover extraction of six different land cover classes in Qiqihar City, a typical black soil region of Northeast China. Therefore, we have replaced the term "land feature extraction" with "land cover extraction" throughout the paper to accurately reflect the focus of the research on land cover extraction.
Point 2:Lines 263-273 clearly state the four objectives of this study, which correspond to Results 3.2, Dicussion4.1, Dicussion.4.2, and Dicussion.4.3, respectively. I would incorporate 4.1-4.3 into Results because they are parts of the results, not extra discussions. Authors may combine all these subtitles into one“Results and Discussion”section.
Response 2:In Section 3.1, the first paragraph describes the parameters used in the experiments conducted in this study, while the second paragraph provides an overall overview of the experiments conducted.Section 3.2 presents the results and analysis of the experiments, focusing on land cover extraction at both a large-scale regional level and a local-scale level. This approach aims to verify the effectiveness and uniqueness of the RAT-UNet model.As for Sections 4.1, 4.2, and 4.3, they are incorporated into the Discussion section. Three sets of comparative experiments were conducted to validate the rationality of the RAT-UNet model construction and the selection of band combinations. Placing these three parts in the Discussion section is considered appropriate, as it allows for a comprehensive examination and validation of the RAT-UNet model's rationale and band combination approach.
Point 3:Figure 3 is not clear, less than 300dpiï¼›Figure 5, Please directly add the names of each land cover type for A-F, and also add a legend for the colorsï¼›Figure 7. What does y axis represent, add the y axis labelï¼›Table 2 and 3. Some land cover names have capitalized initial letter, some don’t have.
Response 3:We have made the necessary adjustments to the figures and tables in the manuscript. The resolution of Figure 3 has been adjusted to 300dpi. We have also provided names for the land cover categories represented by A-F in Figure 5 and added a legend at the bottom of the figure for clarity. In Figure 7, the y-axis now represents extraction accuracy in percentage (%), and the units have been appropriately labeled. Additionally, we have corrected the issue of inconsistent letter sizes in Tables 2 and 3.
Point 4:Section 4.3 and Figure 8. I think the design of these four data combinations doesn’t make any sense. I am more interested in a data combination design for exploring the contribution of visible bands, NIR band, and vegetation indices in land cover type extraction using deep learn model, like b1+b2+b3, b1+b2+b3+b4, and b1+b2+b3+ ndvi+ndwi, and b1+b2+b3+b4+ ndvi+ndwi. b1-b3 represent the basic visible optical bands, the NIR band (b4) and two vegetation indices represent extra or enhanced data information. Also as authors mentioned in Lines 184-200, NIR actually plays important roles in identifying vegetation and water bodies, why the data combination without band 4 (i.e., band 1+band2+band3) obtained highest classification accuracy in vegetation and water body identification (Figure 8). Authors may need to discuss it in the manuscript.
Response 4:In Section 4.3, we discussed the influence of different band combinations on the land cover extraction performance of the RAT-UNet model. Through extensive literature review and analysis, we found that soil organic matter content in the black soil region has the highest correlation with the infrared and near-infrared bands. The band combination of green, red, and near-infrared (b2, b3, b4) yielded the best results for extracting black soil information. We aimed to explore the optimal band combination for land cover extraction in the black soil region. Hence, we compared four different band combinations: blue, green, red (b1, b2, b3), green, red, near-infrared (b2, b3, b4), blue, green, near-infrared (b1, b2, b4), and blue, red, near-infrared (b1, b3, b4). This comparison aimed to provide guidance for future land cover extraction in the black soil region.The band combinations mentioned by the expert, such as b1+b2+b3+b4, b1+b2+b3+ndvi+ndwi, and b1+b2+b3+b4+ndvi+ndwi, are considered to explore the optimal band combinations for specific categories, such as , cropland,water bodies, and forest land. Although these combinations differ slightly from the content expressed in Section 4.3, we appreciate the suggestions made by the expert as they provide valuable insights for our future research.